# The ER membrane protein complex interacts cotranslationally to enable biogenesis of multipass membrane proteins

Matthew J Shurtleff[1†], Daniel N Itzhak[2†], Jeffrey A Hussmann[1], Nicole T Schirle Oakdale[1,3], Elizabeth A Costa[1], Martin Jonikas[1‡], Jimena Weibezahn[1], Katerina D Popova[1], Calvin H Jan[1§], Pavel Sinitcyn[2], Shruthi S Vembar[4], Hilda Hernandez[5], Jürgen Cox[2], Alma L Burlingame[5], Jeffrey L Brodsky[4], Adam Frost[3,6], Georg HH Borner[2]*, Jonathan S Weissman[1,7]*

[1]Department of Cellular and Molecular Pharmacology, University of California, San Francisco, San Francisco, United States; [2]Department of Proteomics and Signal Transduction, Max Planck Institute of Biochemistry, Martinsried, Germany; [3]Department of Biochemistry and Biophysics, University of California, San Francisco, San Francisco, United States; [4]Department of Biological Sciences, University of Pittsburgh, Pittsburgh, United States; [5]Department of Pharmaceutical Chemistry, University of California, San Francisco, San Francisco, United States; [6]Chan Zuckerberg Biohub, San Francisco, United States; [7]Howard Hughes Medical Institute, University of California, San Francisco, San Francisco, United States

*For correspondence:
borner@biochem.mpg.de (GHHB);
Jonathan.Weissman@ucsf.edu
(JSW)

†These authors contributed
equally to this work

Present address: ‡Department
of Molecular Biology, Princeton
University, Princeton, United
States; §Calico Life Sciences LLC,
San Francisco, United States

Competing interests: The
authors declare that no
competing interests exist.

Reviewing editor: David Ron,
University of Cambridge, United
Kingdom

**Abstract** The endoplasmic reticulum (ER) supports biosynthesis of proteins with diverse transmembrane domain (TMD) lengths and hydrophobicity. Features in transmembrane domains such as charged residues in ion channels are often functionally important, but could pose a challenge during cotranslational membrane insertion and folding. Our systematic proteomic approaches in both yeast and human cells revealed that the ER membrane protein complex (EMC) binds to and promotes the biogenesis of a range of multipass transmembrane proteins, with a particular enrichment for transporters. Proximity-specific ribosome profiling demonstrates that the EMC engages clients cotranslationally and immediately following clusters of TMDs enriched for charged residues. The EMC can remain associated after completion of translation, which both protects clients from premature degradation and allows recruitment of substrate-specific and general chaperones. Thus, the EMC broadly enables the biogenesis of multipass transmembrane proteins containing destabilizing features, thereby mitigating the trade-off between function and stability.

DOI: https://doi.org/10.7554/eLife.37018.001

## Introduction

As the primary site of transmembrane protein synthesis, insertion, and folding, the endoplasmic reticulum (ER) must accommodate a diverse range of transmembrane proteins destined for locations throughout the cell. Individual transmembrane domains (TMDs) of multipass proteins are cotranslationally inserted into the lipid bilayer, and this step can often be energetically costly (*Cymer et al., 2015*). Some features of diverse transmembrane proteins present a particular challenge for their insertion into and stabilization within the ER membrane. First, the length of the TMD may not match

the thickness of the lipid bilayer in the ER due to differences in membrane composition between the ER and the protein's final destination (*Sharpe et al., 2010*). Additionally, many TMDs contain features that are destabilizing during membrane protein insertion and biosynthesis but are necessary for function. Transporters, transmembrane ATPases, and solute carriers, for example, contain polar and/or charged residues within membrane spanning domains that form aqueous channels within the plane of the membrane (*Tector and Hartl, 1999*). Yet, during biogenesis, these charged helices enter the lipid bilayer unshielded by the remainder of the protein.

Faced with these challenges, membrane protein folding is subject to failure both in normal and disease settings. Misfolded membrane proteins underlie a host of diseases, including cystic fibrosis (due to mutations in the cystic fibrosis conductance regulator – CFTR), Charcot-Marie-Tooth disease (PMP22 and Connexin 32), diabetes insipidis (Aquaporin), retinitis pigmentosa (Rhodopsin), Niemann-Pick disease (NPC1) (*Gelsthorpe et al., 2008*) and others. Disease-associated mutations often cluster within transmembrane helices (*Sanders and Myers, 2004*) and are biased towards missense mutations, resulting in the addition of polar or charged residues within transmembrane helices (*Partridge et al., 2002*). The disease-associated CFTR mutations illustrate the challenges and opportunities associated with membrane protein biogenesis. Even in healthy cells under normal conditions, less than 50% of newly synthesized CFTR folds properly and traffics out of the ER (*Kopito, 1999*). Small molecule enhancers of folding have emerged as a promising therapeutic strategy for disease-associated misfolded proteins. Indeed, the treatment for the most common form of cystic fibrosis (CFTRΔF508) employs a drug combination including a 'folding corrector' to allow increased ER exit and transport to the cell surface and a 'potentiator' to increase chloride transport activity (*Boyle et al., 2014*). Understanding the full complement of machinery used by the ER to ensure membrane protein synthesis, folding and transport is a fundamental problem with clear biomedical implications.

Although the ER has machinery for stabilizing, sensing and degrading misfolded proteins, how proteins with destabilizing features within transmembrane helices are stabilized during and immediately following synthesis remains poorly understood. The ER membrane protein complex (EMC) has emerged as an intriguing player in membrane protein biogenesis in the ER. The EMC was first described in yeast as a complex of 6 co-purifying and conserved proteins (Emc1-6) with strongly correlated phenotypes in a double mutant genetic modifier map of the unfolded protein response (*Jonikas et al., 2009*). The pattern of EMC genetic interactions strongly resembles the pattern of other factors whose loss leads to the accumulation of misfolded membrane proteins, including the overexpression of a misfolded membrane protein (Sec61-2), and these insights first suggested that the EMC may be a TMD protein chaperone (*Jonikas et al., 2009*). The mammalian EMC orthologues were subsequently identified in a mammalian physical interaction map of ER-associated degradation (ERAD) components (*Christianson et al., 2011*). In vivo experiments have shown that loss of the EMC compromises synthesis, stabilization and/or trafficking of specific multipass membrane proteins in *S. cerevisiae* (a Yor1 mutant which mimicked a common disease allele of CFTR, and Mrh1) (*Louie et al., 2012*; *Bircham et al., 2011*), *D. melanogoster* (rhodopsin) (*Satoh et al., 2015*), *C. elegans* (acetylcholine receptors) (*Richard et al., 2013*) and mice (ABCA3) (*Tang et al., 2017*), and that knockdown of an EMC component compromised biogenesis of mutant CFTR expressed in HeLa cells (*Louie et al., 2012*). In addition, the EMC has been implicated in autophagy (*Shen et al., 2016*; *Li et al., 2013*), lipid transfer and tethering between the ER and mitochondria (*Lahiri et al., 2014*), and flavivirus replication (*Zhang et al., 2016*; *Savidis et al., 2016*; *Marceau et al., 2016*; *Ma et al., 2015*; *Krishnan et al., 2008*). Finally, the EMC was recently shown to act as a posttranslational insertase into the ER membrane for the sterol biosynthesis enzyme, squalene synthase (SQS/FDFT1) and a subset of other tail-anchored (TA) proteins. These TA substrates have moderately hydrophobic TMDs rendering them unable to interact with TRC40/Get3, the cytosolic receptor that delivers TA proteins to the dominant ER insertase, GET1/2 (*24*).

Despite its clear importance, many questions regarding EMC function remain: Is the effect on multipass transmembrane protein biogenesis direct or indirect (e.g. due to changes in lipid composition)? If direct, what is the EMC substrate range and does the EMC physically interact with clients? Lastly, at which stage(s) does the EMC act: insertion into the membrane (*Guna et al., 2018*), co- or post-translationally, during folding, or, finally, during ER export?

Here, we used systematic and unbiased in vivo approaches to identify client proteins and to explore the principles of EMC action with minimal perturbations in both yeast and human cells. Our

studies reveal three conserved principles of EMC function: (1) The EMC interacts with and stabilizes a range of client proteins consisting, with a few exceptions, of multipass transmembrane proteins biased towards transporters. (2) The EMC can initiate client interactions cotranslationally and stabilizes newly synthesized, client proteins after initial ER targeting to prevent premature degradation. (3) The EMC can engage client proteins following clusters of TMDs, and client TMDs are enriched in uncommon transmembrane amino acids (especially charged and bulky residues). Thus, the EMC enables the biogenesis and folding of a subset of multipass membrane proteins which present challenges for the canonical membrane protein synthesis and insertion machineries and thereby expands the functional repertoire of the membrane protein proteome.

## Results

### The EMC physically interacts with multipass transmembrane proteins transiting the ER and substrate-specific chaperones

We initially sought to define the range of proteins that interact with the EMC in the budding yeast *Saccharomyces cerevisae*. To evaluate EMC interaction partners, we endogenously tagged EMC3 with 3xFLAG epitope at its C-terminus. To maximize retention of interacting partners, we recovered Emc3 using a one-step affinity purification in the presence of digitonin. Following SDS-PAGE analysis, prominent bands were excised and analyzed by mass spectrometry to identify Emc3-interacting proteins (*Figure 1A*). As expected, we identified roughly stoichiometric quantities of all core EMC components (Emc1, Emc2, Emc4, Emc5 and Emc6) and the accessory proteins Sop4 and Emc10 (*Jonikas et al., 2009*). Although Sop4 is a near-stoichiometric interactor with the EMC, it shares only a subset of the characteristic genetic interactions of other core components (*Jonikas et al., 2009*), suggesting that, in contrast to the six core EMC components (Emc1-6), the full function of the EMC complex does not depend on Sop4, and it may play a role distinct from the core complex. In addition to EMC components, Emc3 interacted with several large, multi-pass membrane proteins (Pma1, Fks1, Spf1) and Ilm1, a poorly characterized ER-localized protein (*Lockshon et al., 2007*) (*Figure 1A*).

To confirm that these interactions were specific to the EMC, we performed stable isotope labeling with amino acids in cell culture (SILAC) for Emc3-3xFLAG or N-terminally 3xFLAG tagged Orm1 yeast strains. The Orm complex is an unrelated ER resident complex that serves as a control for specific interactions with the EMC. This quantitative analysis verified interactions with all core (Emc1-6) and accessory EMC components (Sop4 and Emc10), and demonstrated specific interactions between the EMC, multipass membrane proteins (Spf1, Fks1, and Pma1) and Ilm1 (*Figure 1B*). We also detected specific interaction with Erg9, the yeast homolog of a TA protein recently shown to be inserted into the ER in an EMC-dependent manner (*Guna et al., 2018*). Therefore, this approach detects both stable and transient interactions between the EMC and binding partners.

### The EMC interacts with specialized membrane protein chaperones

In addition to interacting with putative client transmembrane proteins, our pulldown results implicate the EMC as interacting with membrane protein substrate-specific chaperones, including Sop4 and Gsf2 (*Figure 1B*). Sop4 was previously shown to be a specialized chaperone/transport factor required for the biogenesis of the yeast plasma membrane ATPase (Pma1), (*Luo et al., 2002*), and Gsf2 plays a role in the biogenesis and export of hexose transporter 1 (Hxt1) from the ER (*Sherwood and Carlson, 1999*).

Several considerations suggested to us that the resident ER protein Ilm1 could also act as substrate-specific chaperone for the cell wall synthesis enzyme, Fks1. ILM1 deletion results in enhanced oleic acid sensitivity, likely due to a cell wall defect (*Lockshon et al., 2007*) and showed increased sensitivity to caspofungin, similar to the deletion phenotype for FKS1 (*Markovich et al., 2004*). To explore if Ilm1 functioned as an Fks1 chaperone, we generated yeast expressing Ilm1-3xFLAG and performed pulldowns to identify Ilm1 interacting proteins. We observed a near stoichiometric interaction between Ilm1 and Fks1 (*Figure 1C*). Since mature Fks1 localizes to the plasma membrane and Ilm1 is exclusively in the ER, this likely represents an interaction between Ilm1 and transiting, newly synthesized Fks1. Other prominent bands were identified as components of the EMC (Emc1 and Emc2), ER oxidoreductase 1 (Ero1) and general chaperone proteins (Ssa1, Ssb1 and Kar2). The near

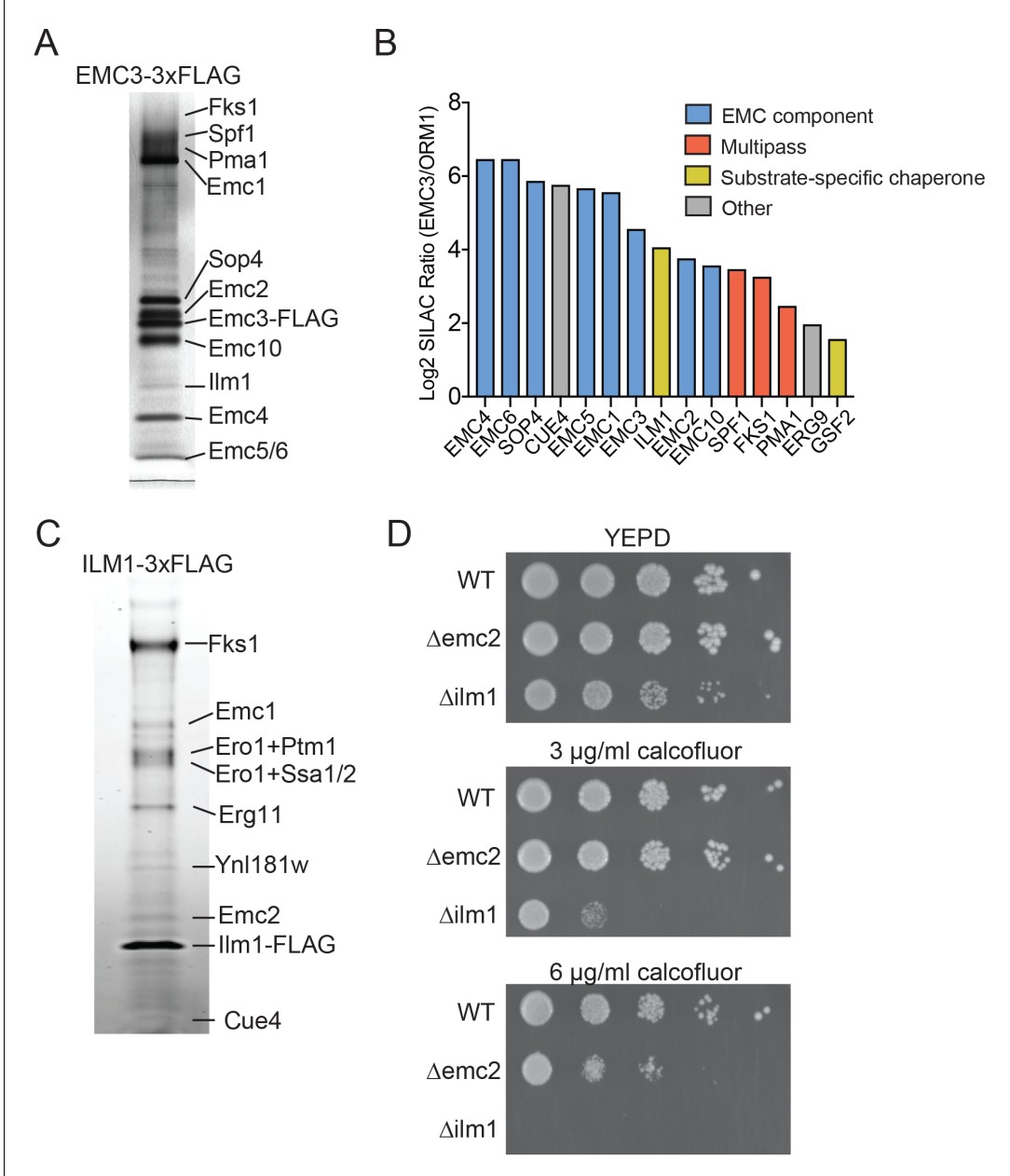

**Figure 1.** Identifying EMC interaction partners in *S. cerevisiae*. (**A**) SDS-PAGE gel of Emc3-3xFLAG co-immunoprecipitated proteins. Proteins identified from excised bands are indicated. (**B**) SILAC ratios for proteins identified by mass spectrometry for Emc3-3xFLAG (heavy) and ORM1-3xFLAG (light) immunoprecipitations. The most Emc3-3xFLAG enriched proteins are shown. (**C**) SDS-PAGE gel of Ilm1-3xFLAG co-immunoprecipitated proteins. Proteins identified from excised bands are indicated. (**D**) 10-fold serial dilutions of log phase cultures of the indicated strains were plated on YEPD, YEPD containing calcofluor white at the indicated concentrations and incubated at 30°C for 24 hr.

DOI: https://doi.org/10.7554/eLife.37018.002

stoichiometric interaction between Ilm1 and Fks1 and interactions with chaperone proteins further suggested a role for Ilm1 as a co-chaperone for Fks1.

As the catalytic subunit of the 1,3-beta-D-glucan synthase cell wall biosynthesis enzyme, FKS1 deletion results in hypersensitivity to compounds, such as calcofluor white, that affect cell wall assembly (*Ram et al., 1995*). If Ilm1 acts as a co-chaperone for Fks1, we hypothesized that a Δilm1 strain would show a calcofluor white hypersensitive phenotype. Indeed, we found profound sensitivity with growth decreased at 3 µg/ml and completely inhibited at 6 µg/ml calcofluor white in the

Δ*ilm1* background (*Figure 1D*). We also noted increased sensitivity compared to wild type for Δ*emc2* at 6 µg/ml, further supporting a role for the EMC in maintaining cell wall integrity, possibly by promoting the biosynthesis of Fks1.

## EMC interacts with transmembrane protein clients independent of substrate specific chaperones

We first hypothesized that the EMC might directly interact with client-specific membrane protein chaperones, which act to bridge the EMC and multipass membrane protein clients. We tested this hypothesis by performing quantitative mass spectrometry of Emc3-3xFlag interacting proteins in wild type, Δ*sop4* or Δ*ilm1* strains (*Figure 2A*). Rather than a decrease in the interaction between the EMC and multipass proteins, as predicted by the bridging model, we observed a prominent increase in EMC association with Fks1 and the functionally redundant paralog, Gsc2, in the Δ*ilm1* background as well as Elo2, Pma1 and Mrh1 in the Δ*sop4* background (*Figure 2B*). Interestingly, Mrh1 was previously shown to be dependent on the EMC for its biosynthesis and cell surface localization (*Bircham et al., 2011*). Consistent with a role for Ilm1 and Sop4 as membrane protein-specific chaperones, we observed a decrease in general chaperone proteins (Ssa1, Ssb1, and Kar2) and an increase in ribosomal proteins associated with EMC3-3xFLAG in strains missing these factors (*Figure 2B*). Together, these pulldown experiments showed that the EMC interacts with multipass membrane proteins transiting through the ER, membrane protein specific co-chaperones, general chaperones, and the ribosome. The presence of client-specific co-chaperones (Ilm1 and Sop4) enhances the interaction between the EMC and general chaperones and decreases the association with both multipass transmembrane clients and the ribosome (*Figure 2C*). These observations

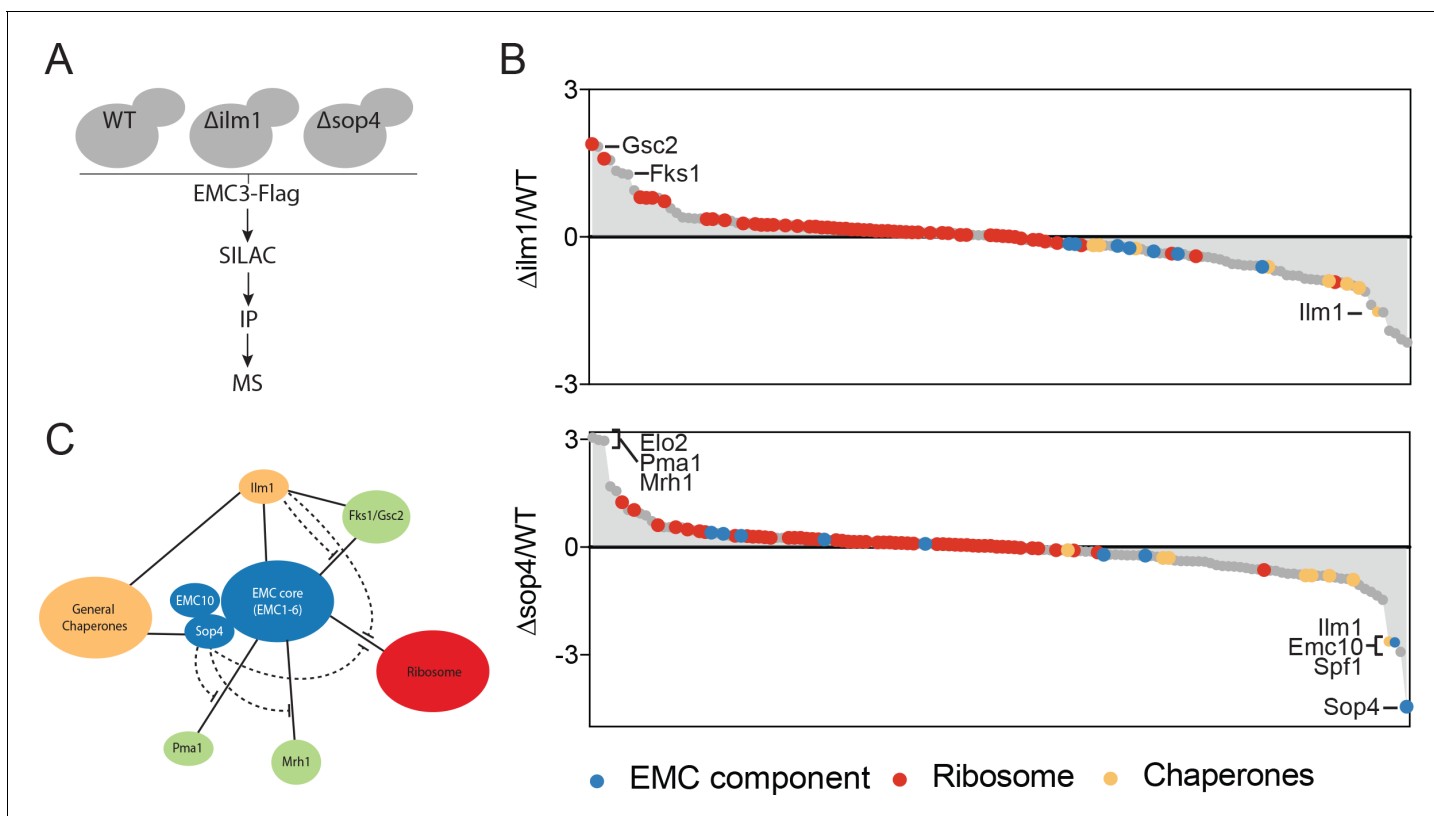

**Figure 2.** The EMC interacts with multipass client proteins independent of co-chaperones. (A) Schematic showing SILAC strategy for comparative analysis of EMC3-3xFLAG interactions in wildtype (WT - light) and Δ*ilm1 (heavy)* and Δ*sop4 (heavy)* cells. IP – immunoprecipitation, MS – mass spectrometry. (B) Log2 SILAC ratios for all proteins identified in EMC3-FLAG expressing strains (top - Δ*sop4* and WT, bottom - Δ*ilm1* and WT). Enriched multipass proteins and strongly depleted proteins are indicated. (C) Schematic showing a summary of physical interactions based on pull downs presented in *Figures 1* and *2*.

DOI: https://doi.org/10.7554/eLife.37018.003

suggest that the EMC interacts directly with multipass transmembrane client proteins early during their synthesis, insertion or folding independent of and possibly prior to chaperone engagement.

## The EMC cotranslationally interacts with folding-challenged multipass client proteins

The observed physical interaction between the EMC and Fks1 as well as the increased association of the EMC with ribosomal proteins following chaperone deletion suggested that the EMC may interact with client proteins cotranslationally. To explore this possibility, we performed proximity-specific ribosome profiling (*Jan et al., 2014*) in yeast expressing a fusion protein of an EMC component with biotin ligase (Emc5-BirA) and ribosomes incorporating an AviTag (the substrate of the biotin ligase). Avi-Tagged ribosomes that contact the EMC are biotinylated upon pulse-labelling with biotin. Following streptavidin-based isolation of the biotinylated ribosomes, deep sequencing of the mRNA fragments protected by affinity-purified ribosomes and comparison to the total pool of ribosomes allows identification of messages translated in the proximity of the EMC (*Figure 3A*). In particular, the ratio of pulldown-to-total footprint reads across a message provides a codon-resolution measurement of when translating ribosomes are most accessible to Emc5-BirA.

The profile of this ratio across FKS1 and GSC2 revealed prominent increases in EMC-ribosome proximity immediately following the translation of clusters of TMDs (*Figure 3B*). This enrichment pattern was not observed in strains with two other ER-localized BirA fusions (BirA-Ubc6-TA and BirA-Ssh1), suggesting that the EMC specifically interacts with these nascent chains shortly after synthesis of TMD clusters by the ribosome. The positional enrichment pattern observed for FKS1 and GSC2 motivated a systematic search for other nascent chains cotranslationally engaged by the EMC. To do this, we computed the ratio of total Emc5-BirA enrichment to total BirA-Ubc6-TA enrichment in a sliding window of 101 codons across all genes We then ranked genes according to the highest enrichment ratio attained anywhere in the gene. In addition to the most enriched genes (FKS1 and GSC2), we identified 51 genes that reproducibly showed localized peaks of Emc5-specific enrichment (defined as being in the top 10% of eligible genes in each of two biological replicates; see Materials and methods) (*Figure 3—figure supplement 1*). This list of putative EMC client proteins includes two genes for which the EMC was previously implicated in their biogenesis (PMA1 and YOR1) (*Louie et al., 2012*; *Luo et al., 2002*). Emc5-BirA positional enrichment for client proteins typically was triggered following synthesis of a cluster of TMDs (*Figure 3—source data 1*). We also performed the same analysis on data from a BirA fusion to Sec63, a component of the Sec translocon, produced for an earlier study (*Aviram et al., 2016*). Intriguingly, patterns of enrichment for Sec63-BirA mirrored some, but not all, of the localized Emc5-BirA peaks (*Figure 3C*).

To gain further insight into the timing of cotranslational engagement of the EMC with client nascent chains, we compared the average enrichments around the first TMD for the various ER localized BirAs (i.e. Emc5, Sec63, Ssh1, and Ubc6-TA). For this analysis, we focused on the set of EMC clients defined above and compared their enrichment to the full set of TMD containing proteins (see TMD annotation in Materials and methods) (*Figure 3D*). Consistent with previous observations (*Jan et al., 2014*), we observed that all of the ER-localized BirAs showed an initial enrichment ~60 codons after the first TMD which likely represents the recruitment of the translating message to the ER by the signal recognition particle (SRP). With the exception of the EMC clients when monitored by Emc5-BirA, this enrichment then levels off or decreases modestly (*Figure 3D*). By contrast, for Emc5-BirA, the EMC clients, but not the full set of TMDs, showed a continued increase in enrichment such that the maximum was achieved following synthesis of an additional ~100 amino acids. These results further indicate that cotranslational EMC association with clients continues after initial targeting, and generally following synthesis of clusters of TMDs (*Figure 3D* and *Figure 3—source data 1*).

Since EMC-specific enrichment peaks followed TMD clusters, we analyzed the number and amino acid composition of TMDs in the 51 putative client proteins. Strikingly, the full list of EMC clients was strongly enriched for multipass proteins (*Figure 4A*) and EMC-interacting TMDs were enriched for charged amino acids and depleted for hydrophobic amino acids, including aliphatic residues common in TMDs (*Figure 4B*). Gene ontology term enrichment on the set of putative EMC clients showed enrichment for terms related to transporters; indeed, a majority of clients are classified as transporters (*Figure 4C*). Notably, integral membrane glycosyltransferases (which must overcome the challenge of transferring hydrophilic sugars onto membrane-associated substrates), including beta-glucan synthase genes (FSK1 and GSC2) defined a second class of putative EMC clients

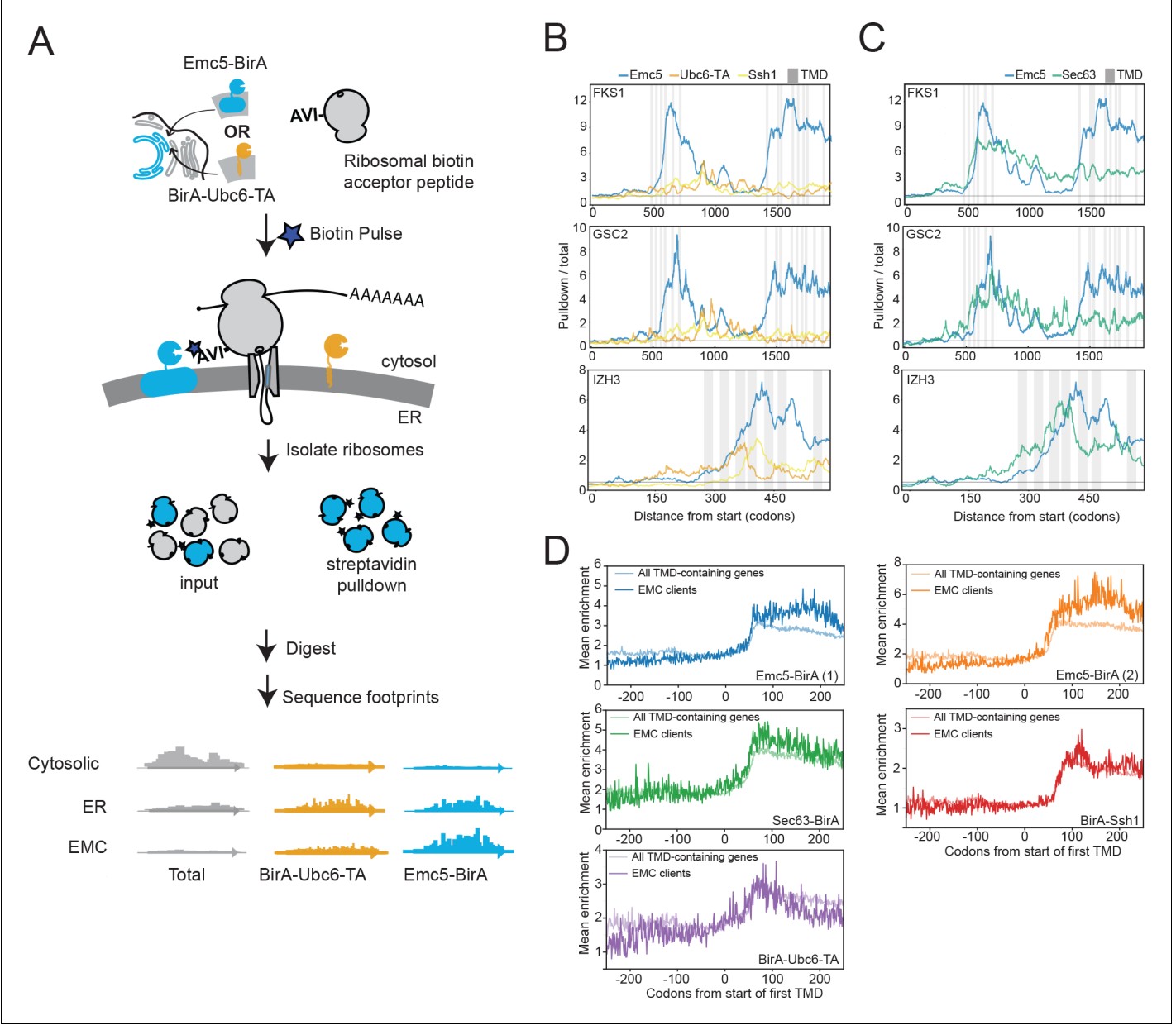

**Figure 3.** The EMC engages client proteins cotranslationally. (**A**) Schematic for strategy to examine the role of the EMC (Emc5-BirA) in cotranslational interaction with clients using proximity-specific ribosome profiling. (**B**) Positional enrichment plots showing footprint reads across the full-length mRNAs of the genes indicated for Emc5-BirA, BirA-Ubc6-TA and Ssh1-BirA expressing strains. Transmembrane domains (TMDs) are indicated in gray. (**C**) As in (**B**), except comparing Emc5-BirA and Sec63-BirA expressing strains. (**D**) Mean enrichments of all TMD-containing genes and EMC clients (N = 51) following start of first TMD for two independent replicates of Emc5-BirA, Sec63-BirA, BirA-Ssh1 and BirA-Ubc6-TA.

DOI: https://doi.org/10.7554/eLife.37018.004

The following source data and figure supplement are available for figure 3:

**Source data 1.** Positional enrichment plots across genes in the >90th percentile for Emc5-BirA/Ubc6-BirA 101 codon window enrichments.
DOI: https://doi.org/10.7554/eLife.37018.006

**Figure supplement 1.** Maximum 101-codon EMC/Ubc6 enrichment ratio windows from two independent Emc5-BirA replicates.
DOI: https://doi.org/10.7554/eLife.37018.005

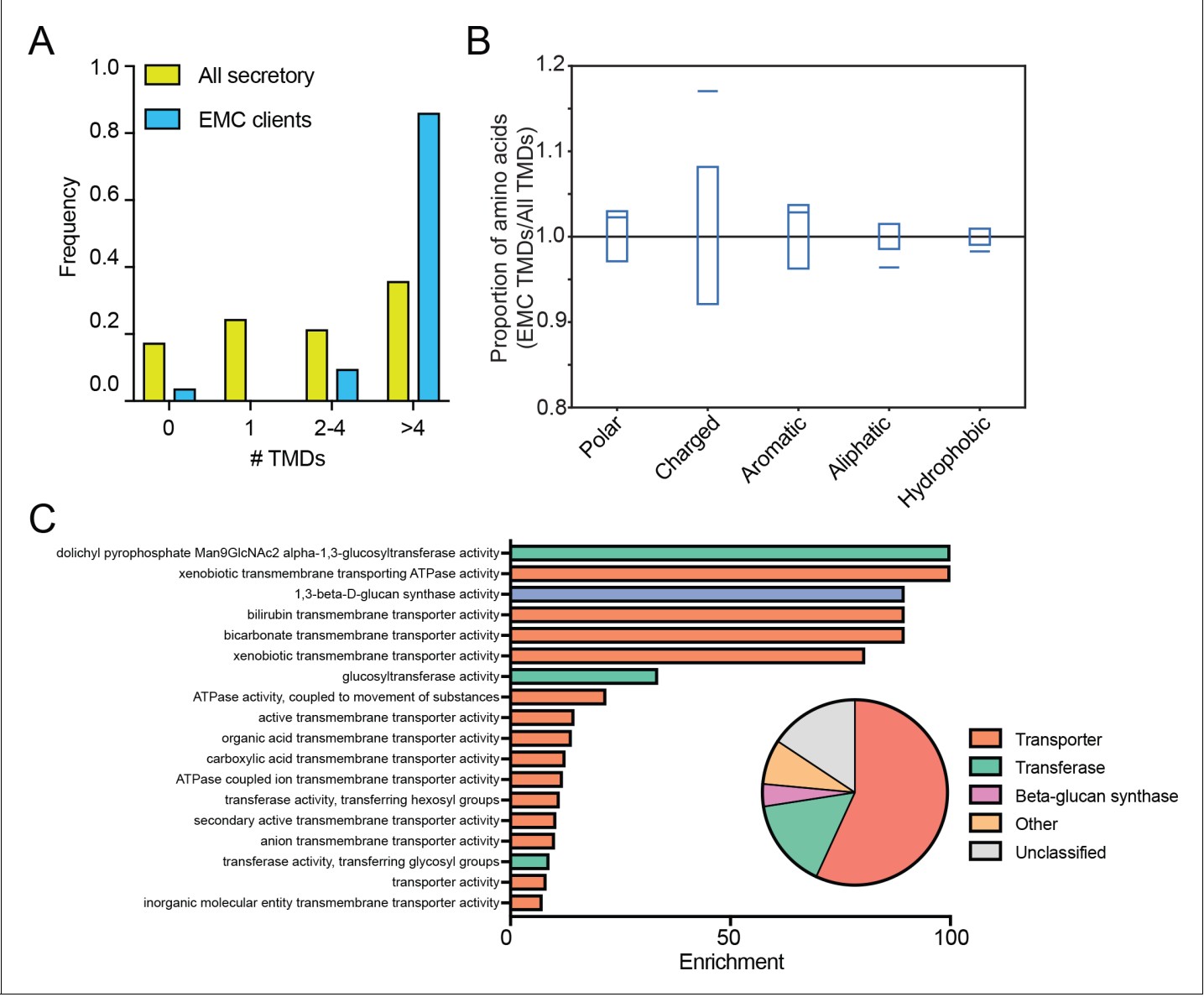

**Figure 4.** EMC clients are enriched for multipass membrane proteins with sub-optimal transmembrane helices. (**A**) Histogram showing the proportion of proteins containing the given number of TMDs for all proteins that enter the secretory pathway (Uniprot annotations) and EMC clients. (**B**) Fraction of amino acids with the given properties in TMDs from EMC client proteins compared to all secretory proteins. Proportion of EMC TMD amino acids/all TMDs for each property is shown by a blue line. Blue boxes indicate 95% confidence ranges defined by 10,000 random sub-samplings of total TMDs with a pool size equal to EMC TMDs (N = 51). (**C**) Top non-redundant over-represented GOMF terms calculated from PANTHER (FDR < 0.05; redundant terms removed by REViGO). Inset: PANTHER protein classifications pie chart for all clients (N = 51).
DOI: https://doi.org/10.7554/eLife.37018.007

(*Figure 4C*). Together, these results suggest that the EMC cotranslationally engages certain multipass membrane proteins and that these clients are enriched for biochemical features and functions that pose a challenge to the general membrane protein biogenesis machinery of the ER.

## Select multipass membrane proteins are EMC clients in mammalian cells

Having identified EMC clients and characterized their properties in yeast, we used an orthogonal approach to identify potential EMC clients in mammalian cells. We reasoned that the EMC may play a role in stabilizing select multipass membrane proteins during synthesis until substrate-specific and general chaperones along with select binding partners, are recruited. This ensemble might then be

required to achieve a structure competent for export or function in the ER. We therefore hypothesized that depletion of the EMC in mammalian cells would result in destabilization and degradation of clients, which could be quantified by global mass spectrometry. Accordingly, we used unbiased quantitative SILAC proteomics to identify putative EMC clients on a proteome-wide scale. Using the CRISPRi system (*Qi et al., 2013*; *Gilbert et al., 2013*), we generated two independent EMC-depleted cell lines in which the expression of either EMC2 or EMC4 was strongly reduced. Whereas EMC2 depletion affects the abundance of all detected members of the EMC complex, depletion of EMC4 has no effect on other EMC members (*Figure 5A*). We thus infer that EMC2 but not EMC4 is structurally integral to the formation of the EMC complex. However, based on the strong similarity in the spectrum of genetic interactions in yeast, *EMC2* and *EMC4* deleted yeast strains, it is likely that the function of the EMC depends on both EMC2 and EMC4 (*9*).

Proteomic changes in these two EMC-depleted cell lines were measured against a control cell line expressing a non-targeting guide RNA (GAL4). Except for EMC components, almost all observed changes were strongly correlated between the EMC2 and EMC4 knockdown cell lines (*Figure 5B*). This supports the notion that depletion of EMC2 and concomitant depletion of the rest of the complex has few additional effects relative to depletion of EMC4 alone, beyond affecting the complex. The identified depleted proteins represent potential EMC client proteins that are destabilized and degraded in the absence of the EMC. To confirm that these clients do not arise from a decreased rate of synthesis, we performed ribosome profiling on EMC2-depleted and control cells, and compared changes in the rate of protein synthesis to changes in steady state proteome abundance (*Figure 5C*). This analysis revealed that the translation rate of the vast majority of potential client proteins is unaffected (with the notable exception of ATP6V0A1, which is decreased at both the translational and protein level), consistent with post-translational degradation. By contrast, proteins that are upregulated at the protein level are also upregulated at the translational level, indicating transcriptional and/or translational induction. As expected from EMC genetic interactions, 2D annotation enrichment (*Cox and Mann, 2012*) shows that proteins upregulated at the translatome and proteome-level are enriched for gene ontology terms related to the unfolded protein response (*Figure 5—figure supplement 1*) (*Jonikas et al., 2009*). By contrast, analysis of the proteins degraded upon EMC-depletion reveals an enrichment for proteins that enter the secretory pathway (*Supplementary File 1*). Moreover, of the 11 proteins decreased by 2-fold or more in both EMC2- and EMC4-depleted cells, 10 have at least one transmembrane domain. Additionally, only 8 of the 37 total hits are soluble proteins with the majority of soluble proteins (5) being localized to the lysosome, and thus likely secondary to depletion of the multipass subunit of the V-ATPase (ATP6V0A1) (*Supplementary file 1*). Confirming a role for the EMC in TA protein insertion (*Guna et al., 2012018*), the levels of mammalian squalene synthase (FDFT1) and TREX1 (*35*) decreased upon EMC depletion. However, similar to that seen in yeast, the cohort of putative EMC client proteins is strongly enriched for multipass transmembrane proteins (*Figure 5D*) and, like the yeast clients, are more likely to contain charged and aromatic residues distributed throughout the TMD (*Figure 5E* and *Figure 5—figure supplement 2*). Further mirroring the yeast EMC client profile, GO terms associated with the identified human clients were enriched for transporter related terms which constituted the largest class of proteins whose abundance decreased upon EMC depletion (*Figure 5F*).

Together, these data support a conserved role for the EMC in facilitating the biogenesis of multipass membrane proteins with destabilizing membrane spanning sequences. Depletion of the EMC renders some client proteins unable to attain normal expression levels.

## Discussion

Our work establishes that the predominant function of the EMC is to ensure the biogenesis of a subset of multipass membrane proteins. These studies are consistent with a model in which the EMC cotranslationally interacts with nascent polypeptides. Following synthesis, the EMC stabilizes clients and enables recruitment of substrate-specific and general chaperones to achieve a conformation that is competent for function in the ER or for transport to the Golgi (*Figure 6*). In the absence of the EMC, newly synthesized client proteins are likely extracted from the membrane for degradation by the UPS (*Figure 6*).

Our studies reveal three principles of the EMC's action. First, the EMC directly interacts with and stabilizes a range of client proteins consisting primarily of multipass transmembrane proteins. Several

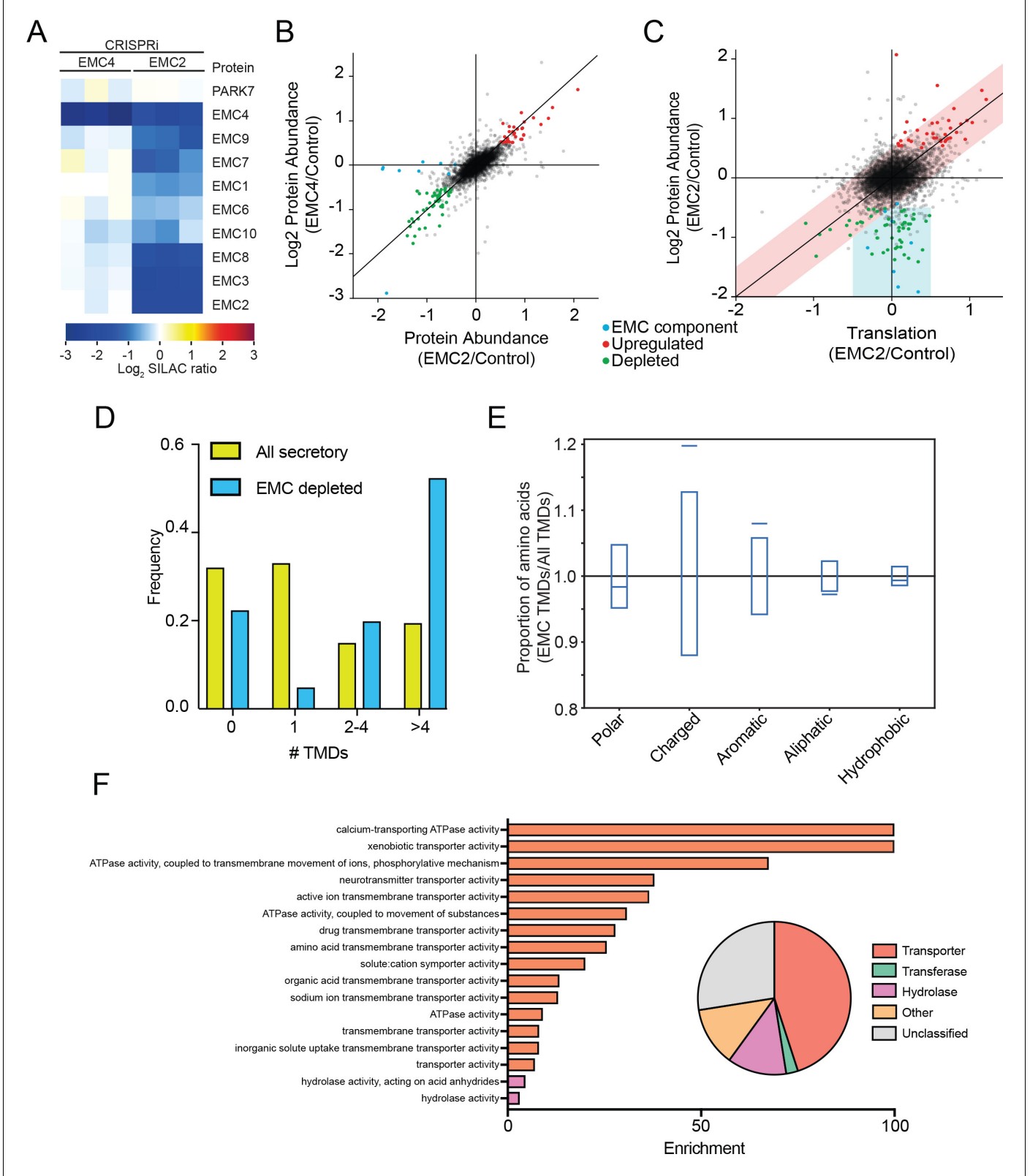

**Figure 5.** The mammalian EMC stabilizes multipass transmembrane proteins. (**A**) SILAC quantification of EMC components, comparing their expression level in cells with guide RNAs targeting EMC2 or EMC4 with those expressing non-targeting GAL4 guide RNA. PARK7 is shown as a control (**Wiśniewski and Mann, 2016**). (**B**) Full proteome comparison scatter plot of protein abundance change in cells expressing EMC2 guide RNA against abundance change in cells expressing EMC4 guide. Expression is relative to non-targeting GAL4 guide RNA. Proteins colored red are significantly

*Figure 5 continued on next page*

*Figure 5 continued*

upregulated in both EMC2 and EMC4 cells. Proteins colored green are significantly downregulated in both EMC2 and EMC4 cells (Log2 >0.5). EMC components are colored blue. (**C**) Comparison of translation change by ribosome profiling with proteome change. (**D**) Histogram showing the proportion of proteins containing the given number of TMDs for all proteins that enter the secretory pathway (as defined in Uniprot) and EMC clients. (**E**) Fraction of amino acids with the given properties in TMDs from EMC client proteins compared to all secretory proteins. Proportion of EMC TMD amino acids/all TMDs for each property is shown by a blue line. Blue boxes indicate 95% confidence ranges defined by 10,000 random sub-samplings of total TMDs with a pool size equal to EMC TMDs (N = 37). (**F**) Top non-redundant over-represented GOMF terms calculated from PANTHER (FDR < 0.05; redundant terms removed by REViGO). Inset: Protein classifications pie chart for EMC client proteins (N = 37).

DOI: https://doi.org/10.7554/eLife.37018.008

The following figure supplements are available for figure 5:

**Figure supplement 1.** 2D annotation enrichment based on the protein ratios and ribosome profiling ratios of EMC knockdown versus control knockdown cells.

DOI: https://doi.org/10.7554/eLife.37018.009

**Figure supplement 2.** Amino acid composition of transmembrane domains of EMC clients and background.

DOI: https://doi.org/10.7554/eLife.37018.010

observations support this conclusion. The EMC physically associates with multipass proteins in yeast (*Figure 1 A and B*). Deletion of an EMC component and dedicated membrane protein chaperone (SOP4) results in increased interaction with several multipass membrane proteins that transit the ER, including a previously identified Sop4 substrate (Pma1) (*Luo et al., 2002*) and a membrane protein

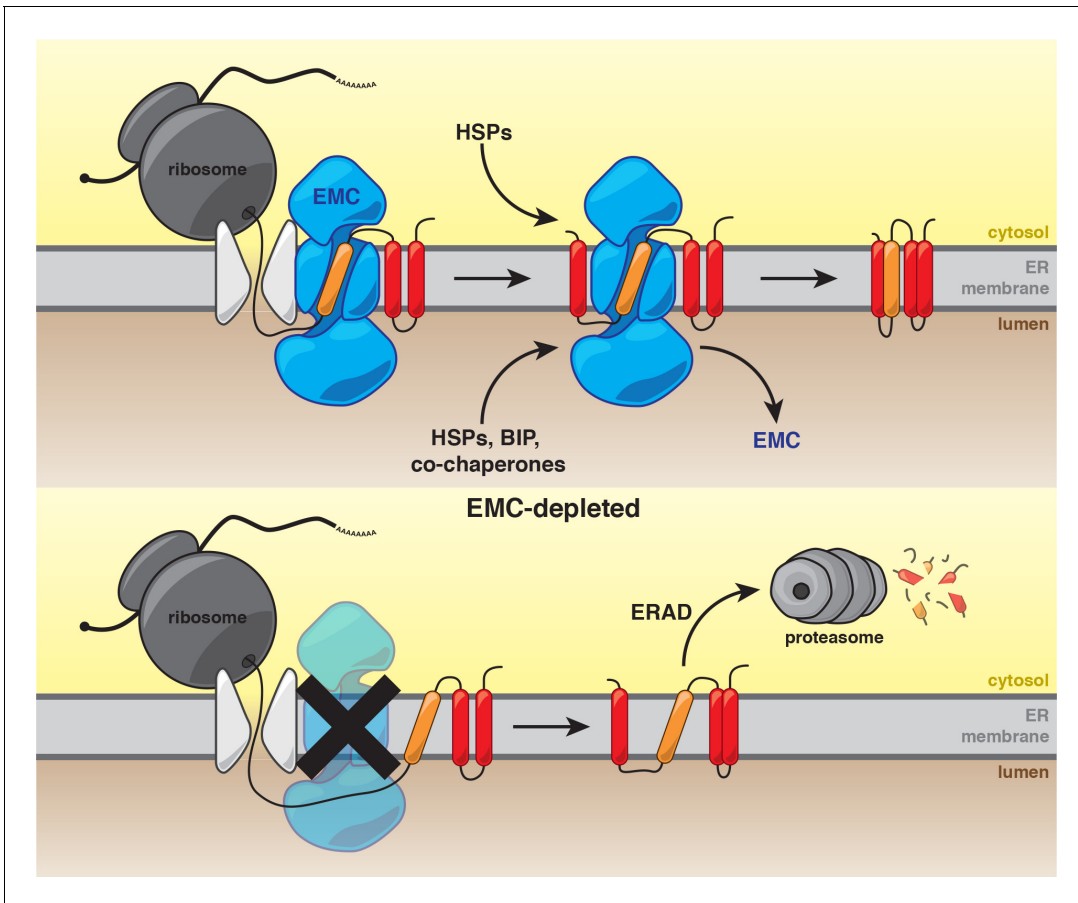

**Figure 6.** Model for the role of the EMC in multipass transmembrane protein biogenesis. See text for details. Unstable transmembrane domains are shown in orange. Note, while the EMC is depicted here as cooperating with the translocon following insertion, our data do not exclude the possibility that the EMC acts as an insertase for some substrates.

DOI: https://doi.org/10.7554/eLife.37018.011

previously shown to require the EMC to localize to the cell surface (Mrh1) (*Bircham et al., 2011*) (*Figure 2B*). In addition, we show that the poorly characterized ER resident protein Ilm1 acts as a substrate-specific chaperone for Fks1 (*Figure 1 C and D*), and ILM1 deletion results in increased interactions between the EMC and Fks1 (*Figure 2B*). These yeast studies show that the EMC directly interacts with multipass protein clients early in biosynthesis independent of, and likely prior to, engagement by their dedicated chaperones. Multipass proteins are also selectively destabilized in human cells depleted of the EMC by CRISPRi (*Figure 5*), further demonstrating that the biogenesis/chaperoning of multipass membrane proteins is a conserved feature of EMC function.

Second, the EMC can begin to interact cotranslationally and subsequently stabilize newly synthesized client proteins, likely preventing premature degradation by ERAD. Proximity-specific ribosome profiling in yeast revealed that a common feature of the EMC is cotranslational engagement of multipass client proteins (*Figure 3*). The presence of full-length multipass membrane proteins in our pulldown analyses in yeast (*Figures 1* and *2*) indicate that the EMC can engage clients cotranslationally and remain bound after completion of protein synthesis for at least a subset of client proteins. Although the EMC was recently shown to act as a TA protein insertase (*Guna et al., 2018*), our results suggest that the EMC interacts with multipass proteins during synthesis and remains associated post-ER targeting (*Figure 3D*). In addition, all of the EMC clients identified in yeast are dependent on SRP, a factor that cotranslationally delivers substrates to the ER surface, and are cotranslationally targeted to the ER (*Costa et al., 2018*). Similarly, all of the mammalian multipass protein EMC substrates were previously found to be cotranslationally targeted to the ER by proximity-specific ribosome profiling studies (*Jan et al., 2014*). These results, however, do not exclude the possibility that the EMC acts to insert or facilitate flipping of individual helices of multipass membranes proteins following initial targeting to the ER, or that the EMC acts during the initial insertion for a subset of the newly identified client proteins. Overall, our results support a model in which the EMC can engage client proteins cotranslationally, perhaps following insertion via the translocon into the ER membrane, likely acting to stabilize folding intermediates and to forestall degradation.

Third, the EMC engages transporters and other client proteins enriched for sub-optimal residues in transmembrane helices. Our proximity-specific ribosome profiling results indicate that the EMC typically engages transporters and other membrane proteins enriched for charged residues in TMDs (*Figure 4 B and C*) following synthesis of TMD clusters (*Figure 3B*). In addition, membrane proteins that were destabilized in human cells were also enriched for transporters (*Figure 5F*), and human EMC clients were enriched for charged and bulky residues in TMDs (*Figure 5E*). These observed features of EMC clients are inter-related, multipass membrane proteins are more likely to have sub-optimal helices and are enriched for transporter related functions. Indeed, many transporters contain polar and charged residues within TMDs, which are necessary for solute delivery across the lipid bilayer. While our studies in HeLa cells primarily identified solute carriers and transmembrane ATPases as clients, the EMC was previously implicated in the biogenesis of cell-type specific multimeric channel proteins that also contain TMDs with charged residues (*Richard et al., 2013*). Thus, we propose that the EMC promotes the biogenesis of a wide range of human multipass client proteins by stabilizing transmembrane regions during biosynthesis and prior to completion of folding.

How does the EMC act both as a cotranslational TMD chaperone, following ER targeting, and as a post-translational insertase for a subset of TA proteins (*Guna et al., 2018*)? We suggest that these phenomena reflect a common biochemical property of the EMC: the ability to interact with transmembrane helices with low hydrophobicity. This property is consistent with the previously reported role for the TMD of EMC1 in binding to and stabilizing a destabilized, ER membrane embedded SV40 virion (*Bagchi et al., 2016*). Therefore, we propose that a unifying function of the EMC is to accommodate and stabilize the wide diversity of membrane spanning sequences by directly interacting with select membrane proteins with destabilizing features in TMDs. Interestingly, EMC3 may share a common ancestry with the universally conserved YidC/Oxa1/Alb3 protein family in bacteria and mitochondria (*Anghel et al., 2017*). YidC also plays a dual role in both the insertion of membrane proteins (*Samuelson et al., 2000*) and the stabilization of membrane proteins inserted via the translocon by direct interaction with SecYEG (*Nagamori et al., 2004*).

Beyond its role in engaging atypical TMDs, our studies point to a broader role for the EMC as a nexus of TMD folding, which requires chaperone recruitment and protection from the ERAD machinery. The EMC is a large protein complex with significant predicted mass integral to the membrane, as well as soluble mass in the cytosol and lumen. It is counterintuitive that such mass is necessary to

act solely as a transmembrane chaperone and insertase. For example, members of the YidC/Oxa1/Alb3 membrane protein family in bacteria and mitochondria act as insertases and transmembrane chaperones, but they function as smaller monomers without appended soluble domains (*Nagamori et al., 2004*). Indeed, we find that in yeast the EMC also recruits substrate-specific chaperones including Ilm1, Sop4 and Gsf2, as well as general cytoplasmic and lumenal chaperones and oxidoreductases. We propose that, in analogy to the recently described function of the Slp1-Emp65 complex for soluble lumenal ER proteins (*Zhang et al., 2017*), the EMC may also hold folding intermediates in a privileged, ERAD-protected state until the recruitment of substrate-specific chaperones and/or general chaperones to complete folding and allow ER export. Thus, the EMC may compartmentalize multiple functions necessary for membrane protein biogenesis: transmembrane stabilization, recruitment of folding factors and protection of folding intermediates from recognition and degradation by the ER quality control machinery.

## Materials and methods

### Yeast strains and plasmids

Strains BY4741 (*MAT*a *his3Δ1 leu2Δ0 met15Δ0 ura3Δ0*) and W303 (*ade2-1 leu2-3 his3-11,15 trp1-1 ura3-1 can1-100*) were used as wild-type parental strains. Genomic knockouts and knockins were generated by one-step gene replacement as described (*Rothstein, 1991*). Generation of EMC3-3X-FLAG (BY4741: EMC3-FLAG-Nat[R]) was described previously (*Jonikas et al., 2009*). To generate FLAG-tagged Ilm1p, the *ILM1* coding sequence including the open reading frame and ~350 base pairs of the 3'-UTR was amplified from genomic DNA extracted from BY4742 wild-type yeast. The resulting PCR products were inserted into a plasmid immediately upstream of an in-frame fusion with 3X-FLAG and the *NATMX6* coding sequence. Next, a 3XFLAG epitope was introduced at the 3' end of the *ILM1* open reading frame in pILM1-UTR-NAT using a modified site-directed mutagenesis protocol (*Wang and Malcolm, 1999*). For homologous recombination, the resulting *ILM1-3XFLAG-UTR-NAT* sequence was amplified and transformed into BY4741. Positive clones were selected on yeast extract-peptone-dextrose (YEPD) medium supplemented with nourseothricin and analyzed for FLAG-tagged Ilm1p expression by immunoblotting with anti-FLAG antibodies (Santa Cruz Biotechnology).

### Immunoprecipitation from microsomes

Immunoprecipitations of Emc3-3xFLAG and Ilm1-3xFLAG were performed as described previously (*Denic and Weissman, 2007*). Yeast were grown in 3L YEPD, harvested and resuspendend in 2 ml lysis buffer (50 mM HEPES-KOH pH 6.8). Resuspended yeast pellets were frozen dropwise in liquid nitrogen and subsequently disrupted by bead beating. 15 ml lysis buffer was added to the frozen yeast powder lysis was performed by 10 strokes in a Dounce homogenizer. The homogenate was centrifuged at 1000Xg for 10 min at 4°C. The supernatant was transferred to a new tube and the centrifugation was repeated. The supernatant was then transferred to a 50.2 Ti ultracentrifuge tube (Beckman Coulter) and centrifuged at 22,000 RPM for 16 min at 4°C. The microsome pellet was resuspended in 0.5 ml lysis buffer, flash frozen in liquid nitrogen and stored at −80°C until use.

Microsomes were solubilized in 15 ml immunoprecipitation buffer (100 mM HEPES-KOH pH 6.8, 300 mM KOAc, 4 mM MgOAc, 2 mM CaCl$_2$, 30% glycerol) for 1 hr at 4°C. Detergent extracted microsomes were centrifuged for at 22,000 RPM for 16 min at 4°C. 150 µl of anti-FLAG bead slurry was added the supernatant and incubated for 2 hr at 4°C. Beads were pelleted at 1,000Xg for 1 min and washed 4 times in 10 ml wash buffer (immunoprecipitation buffer with 0.1% digitonin). Beads were eluted with 150 µl of 2 mg/ml FLAG peptide in was buffer. Eluates were stored at −80°C until use.

### In-gel tryptic digestion and mass spectrometry

Immunoprecipitation eluates were separated by SDS-PAGE and single bands were excised, or alternatively, gels were cut along the lanes in 11 to 13 pieces. Proteins were subjected to in-gel digestion (University of California San Francisco Mass Spectrometry Facility protocol) with trypsin (porcine, side-chain protected, Promega). The extracted digests were vacuum evaporated and resuspended

in 0.1% formic acid in water. The digests were analyzed by capillary HPLC-tandem mass spectrometry. The separation was performed with a $C_{18}$ PepMap 75 μm × 150 mm column (LC Packings, Sunnyvale, CA) used on either an Ultimate HPLC system linked with a FAMOS autosampler (LC Packings, San Francisco, CA) or an Agilent 1100 series HPLC system equipped with an autosampler (Agilent Technologies, Palo Alto, CA). The column effluent was directed to either a QSTAR-Pulsar or QSTAR-Elite tandem mass spectrometer (Applied Biosystems/MDS Sciex, Toronto, Canada). Throughout the chromatographic separation, a 1 s MS acquisition was followed by a 3 s CID acquisition for computer-selected precursor ions in information-dependent acquisition mode. The collision energy was set according to the *m/z* value and charge state of the precursor ion.

Data was analyzed with Analyst QS 1.1 software (Applied Biosystems/MDS Sciex) and peak lists were generated using the mascot.dll script (Mascot.dll 1.6b18, Applied Biosystems). Precursor mass tolerance for grouping was set to 0.2 amu. MS centroiding parameters were 50% peak height and 0.02 amu merge distance. MS/MS centroiding parameters were 50% peak height and 0.05 amu merge distance.

The peak lists were searched in in-house Protein Prospector version 5.3.0 (a public version is available on line). Peptides containing one miscleavage were allowed. The number of modifications was limited to two per peptide. Carbamidomethylation modification of cysteine; acetylation of the N terminus of the protein; oxidation of methionine; and formation of pyro-Glu from N-term Gln were allowed as variable modifications. Mass tolerance for was 150 ppm for precursor and 0.2 Da for fragment ions.

## Proximity-Specific ribosome profiling

Proximity-based ribosome profiling was performed essentially as previously described (*McGlincy and Ingolia, 2017*).

### Strain construction

The endogenous copies of RPL10a were C-terminally tagged with an engineered HA-TEV-AviTag sequence to allow for detection by western blot, biotinylation, and specific elution after streptavidin pulldown via TEV protease cleavage. BirA fusion proteins EMC5 and SSH1 were endogenously tagged at the C-terminus (EMC5) or N-terminus (SSH1), respectively with a BirA (biotin ligase), allowing the specific biotinylation and streptavidin pull-down of ribosomes in close proximity to the EMC specifically or to the ER. Generation of Sec63-Bir was previously described (*Jan et al., 2014*).

### Media and growth conditions

Yeast were grown in biotin-free, synthetic defined media (1.7 g/L YNB-Biotin [Sunrise Science Products], 5 g/L Ammonium sulfate, 20 g/L dextrose, complete amino acids) supplemented with d-biotin (Sigma) to a final concentration of 0.125 ng/mL, at 30°C with vigorous shaking. Twenty milliliters of an overnight culture was used to inoculate a 300 ml culture at an OD600 of 0.05–0.1, and biotin induction was performed at mid-log phase with an OD600 of 0.5–0.6 as in (*Jan et al., 2014*).

### Biotin induction and harvesting

Cyclohexamide (CHX) was added to media 2 min prior to the addition of biotin, at a final concentration of 100 μg/mL. To induce biotinylation, D-biotin was added to the media to a final concentration of 10 nM and biotinylation was allowed to proceed for 2 min at 30°C while shaking. After 2 min, cells were harvested by filtration onto 0.45 μm pore size nitrocellulose filters (Whatman), scraped from the membrane, and immediately submerged in liquid nitrogen.

For western-blot quantification, 1 mL aliquots were taken from uninduced cultures and placed into pre-chilled, 1.5 mL siliconized microcentrifuge tubes. Samples were then spun at 20,000 x g at 4°C for 30 s, the supernatant removed, and the pellet-containing tubes immediately placed in liquid nitrogen. For levels of biotin induction quantification, a small patch of induced, filtered cells were scraped from the nitrocellulose filters.

## Western blotting and biotinylation quantification

Lysates were prepared from pelleted induced and uninduced yeast by resuspending frozen pellets in 30–50 µL Laemmli buffer, followed by denaturation at 95°C for 5 min, and clarification at room temperature by spinning at 20,000 x g for 10 min.

Lysates were run on 4–12% Bis-tris gels in MOPS buffer, transferred to nitrocellulose membrane using the BioRad Transfer system (BioRad) according to the manufacturer's instructions, blocked with Odyssey blocking buffer, and subsequently probed. The HA epitope tag was detected using a mouse a high-affinity rat anti-HA antibody at a 1:1000 dilution (Roche 3F10). IR800 anti-rat (Rockland) secondary antibody was then used at 1:5000 dilution. Biotin was detected directly using Streptavidin AlexaFluor 680 (Molecular Probes) at a 1:5000 dilution in TBS-T and a 10 min incubation period after incubation in secondary antibody. All blots were visualized using the Licor (Odyssey) system.

Percent biotinylation was quantified by probing for HA in a streptavidin shift assay, in which clarified lysates were mixed with excess unlabeled streptavidin (Rockland) prior to electrophoresis and immunoblotting. Biotinylated AviTags shift to a higher molecular weight than the corresponding, non-biotinylated AviTags, and percent biotinylation was quantified from the fraction of total signal that was shifted (Algire et al., 2002).

## Yeast lysis, lysate desalting and monosome isolation

650 µL of polysome lysis buffer (20 mM Tris pH 8.0, 140 mM KCl, 1.5 mM MgCl2, 100 µg/mL CHX, 1% Triton X-100) was dripped into a 50 mL conical tube filled with and immersed in liquid nitrogen, containing the harvested yeast strip/pellet from a mid-log phase 300 mL biotin-induced culture. The frozen cell-buffer mixture was cryogenically pulverized for a minute in a freezer mill. Pulverized cells were thawed and centrifuged for 2 min at 4°C and 20,000 x g. The supernatant was immediately loaded onto pre-chilled, 2 mL Zeba de-salt spin column previously equilibrated with polysome gradient buffer (20 mM Tris pH 8.0, 140 mM KCl, 5 mM MgCl2, 100 µg/mL CHX, 0.5 mM DTT) according to the manufacturer's instructions. Aliquots of this extract were flash-frozen in liquid nitrogen and stored at −80°C, typically yielding 0.5–1 mL of extract with A260 of 100–300. A 200–500 µL aliquot of the above lysate was treated with 750 U RNaseI (Ambion) per 50 A260 units of lysate (or 15 U RNaseI per 40 µg RNA, where 1 A260 unit corresponds to 40 µg RNA), and incubated for 1 hr at room temperature on an overhead roller. Reactions were then quenched with 10 µL SUPERase-In RNase inhibitor (Ambion) and stored on ice until loaded onto sucrose density gradients (10–50% w/v) prepared with the polysome gradient buffer described above. Gradients were made in Sw-41 ultracentrifuge tubes (Seton Scientific) using a BioComp Gradient Master (BioComp Instruments) according to the manufacturer's instructions. Samples were spun for 3 hr at 4°C and 35,000 rpm in an Sw-41 rotor (Beckmann Coulter). Fractionation was performed on the Gradient Master using a BioRad EM-1 Econo UV monitor to continually monitored A260 values. Monosome peaks were collected, flash-frozen in liquid nitrogen, and stored at −80°C. Typical yields were 2–3 mL of monosomes with A260 of 2–5.

## Streptavidin pulldown of biotinylated ribosomes

Biotinylated ribosomes were isolated from the total monosome fraction using MyOne streptavidin C1 magnetic DynaBeads (Invitrogen). The volume of beads used per pulldown was scaled based on 187 µL (1.87 mg) beads per 15 pmol of biotinylated ribosomes, as estimated from the manufacturer's instructions. The pmol of biotinylated ribosomes in a given volume was calculated from (i) the fraction of biotinylated ribosomes as estimated from a streptavidin shift assay and (ii) the total concentration of 80S ribosomes in the fraction (Jan et al., 2014). Prior to binding, beads were washed twice with one volume (equal to the initial bead volume) of Buffer A (100 mM NaOH, 50 mM NaCl), once with one volume of Buffer B (100 mM NaCl), and once with one volume of low-salt binding Buffer C (20 mM Tris pH 8.0, 140 mM KCl, 5 mM MgCl2, 100 µg/mL CHX, 0.5 mM DTT, 0.1% Triton X-100). Triton X-100 was added to monosome fractions containing 15 pmol of biotinylated ribosomes, to a final concentration of 0.01%. This solution was added to washed beads, mixed well, and the pulldown was allowed to proceed on an overhead roller for 1 hr at 4°C. The supernatant was removed and the beads were washed three times with 1 mL high-salt wash Buffer D (20 mM Tris pH 8.0, 500 mM KCl, 5 mM MgCl2, 100 µg/mL CHX, 0.5 mM DTT, 0.1% Triton X-100), each for 20 min

at 4°C. After the third wash, beads were re-equilibrated in low-salt Buffer C by resuspension in 1 mL, then transferred to a new tube and resuspended in a smaller volume (200 µL) of Buffer C in preparation for elution by TEV protease cleavage. Cleavage was performed by incubation on a nutator with in-house TEV protease for 1 hr at room temperature. Three volumes of Trizol LS (Ambion) were added to both the TEV eluate and a separate, matched input sample consisting of 10–20 pmol of total monosomes.

## Library generation

Ten-twenty pmol of monosomes in Trizol LS were extracted using 200 µL chloroform per 750 µL Trizol LS. RNA was precipitated for at least 1 hr at −30°C (or 30 mins at −80) using GlycoBlue (Invitrogen) and an equal volume of isopropanol, pelleted, resuspended in 11 µL (input) or 5 ul (pulldown) 10 mM Tris pH 7.0, and resolved on a 15% TBE-urea gel. Samples were denatured in 2X TBE-Urea loading buffer at 80°C for 2 min. Gel was run at 200V for 60 min and visualized after 5 min incubation with SYBR Gold (Invitrogen). Oligoribonucleotide size standard in neighboring lanes was used to excise roughly 28 nt ribosome footprints. Footprints were passively eluted on a tube nutator overnight at 4°C in 420 µL 0.3 M NaCl after crushing gel slices. After overnight RNA elution from gel, ribosome footprints were then precipitated with GlycoBlue and 2.5 volumes ethanol, resuspended directly in a dephosphorylation master mix containing 8 µL 1.25x T4 polynucleotide kinase (PNK) buffer (New England Biolabs, NEB), and dephosphorylated with 2 µL PNK for 1 hr at 37°C. This solution was used directly for ligation to 0.5 µg 3' miRNA cloning linker 1 (Integrated DNA Technologies) upon addition of 8 µL 50% PEG (NEB), 1 µL 10x truncated T4 RNA ligase 2 K227Q (rnl2) buffer (NEB), and in-house rnl2 enzyme as in (**Kopito, 1999**) (or 3.5 ul 50% PEG, 0.5 ul µL 10x T4 RNA ligase buffer, 0.5ul T4 RNA ligase rnl2, and 0.05 ul 1M DTT, and a sample-specific barcode linker as in [**McGlincy and Ingolia, 2017**]). Ligation proceeded for 3 hr at 25°C at which point RNA was precipitated for at least 1 hr at −30°C, purified on a 10% TBE-urea gel, eluted, and precipitated as above.

rRNA contaminants were removed from ligation products using antisense biotinylated oligos as described (**Brar et al., 2012**). rRNA-depleted ligation products were then reverse-transcribed in a 16.7 µL reaction using SuperScript III (Invitrogen) for 30 min at 48°C. RNA template was hydrolyzed for 20 min at 98°C after addition of 1/10 vol 1 M NaOH. Equi-molar HCl was added to quench the reaction and cDNAs were precipitated at −30°C for at least 1 hr and subsequently purified on a 10% TBE-urea gel, eluted overnight, precipitated, and resuspended in 15 µL nuclease-free water.

cDNAs were circularized using CircLigase (Epicentre) in a 20 µL reaction for 1.5 hr at 60°C according to the manufacturer's instructions. Circularized products were amplified by 8–16 cycles of PCR using oNTI231 and any of several Illumina indexing primers (IDT) using Phusion polymerase (Finnzymes) in a 17 µL reaction. PCR amplicons were gel purified on 8% non-denaturing TBE gels, eluted, precipitated, resuspended in 10 µL EB, and quantified using the Bioanalyzer High Sensitivity DNA assay (Agilent Technologies). 2 nM dilutions were multiplexed as needed and sequenced via a single-end run on an Illumina HiSeq sequencer.

## Identification of co-translationally engaged regions

Sequencing reads were trimmed of adaptor sequences and aligned to yeast coding sequences as previously described (**Hussmann et al., 2015**). To compute the efficiency with which ribosomes translating a particular region of a coding sequence were labeled by a BirA fusion, for each codon position in each coding sequence, for both pulldown and input samples, we calculated the sum of ribosome profiling reads in a 50 codon window on either side (i.e. 101 total codons) of the position normalized to the total number of mapped reads for the sample, then computed the enrichment ratio of pulldown reads to input reads in the window. To quantify the extent to which a gene has any region for which ribosomes translating that region are more efficiently labeled by Emc5-fused BirA than a BirA fused with the tail-anchor from Ubc6 (BirA-Ubc6-TA), we computed the maximum value of the ratio of (pulldown/total enrichment for Emc5-BirA) to (pulldown/total enrichment for BirA-Ubc6-TA) for each position in the gene.

The following filters were applied to restrict the set of relevant genes:

- Uniquely mappable: to exclude artifacts from genes that are too similar in sequence, genes were required to have >80% of positions in the coding sequence uniquely mappable when tested with synthetic 25 nt reads.
- Expression cutoff: to exclude noise from very lowly expressed genes, genes were required to have higher than 0.02 reads per codon per million reads in the input (non-pulldown) ribosome profiling for every BirA, and to have no 101 codon window in which there were 0 total reads in any ribosome profiling sample.
- Localization: to exclude noise from genes that are not translated in the proximity of any BirA at all, genes were required to have RPKM(pulldown)/RPKM(input)>1 for at least one BirA ribosome profiling sample pair.

Amongst genes passing these filters, we identified potential EMC clients as those genes whose maximum positional enrichment ratio was in the top 10% of all genes in both biological replicates of the Emc5-BirA ribosome profiling.

## Transmembrane domain annotations

Annotated transmembrane domains were collected from two sources: domains predicted by TMHMM in yeast were downloaded from SGD (date stamp on file: 8/23/2017), and domains annotated in UniProt for yeast and human were extracted from Uniprot (Reviewed Swiss-Prot) databases in xml format (file names uniprot-reviewed%3Ayes + taxonomy%3A4932.xml for yeast and uniprot-reviewed%3Ayes + taxonomy%3A9606.xml for human). For yeast, the sets of domains from both sources were merged, and when a TMHMM prediction overlapped with a domain in Uniprot, the Uniprot domain was chosen. Domains in mitochondrially encoded genes, dubious ORFs, and pseudogenes were excluded. Final sets of transmembrane domains considered are shown in **Supplementary file 2**.

## Amino acid composition of TMDs

To evaluate biochemical properties of TMDs in potential EMC clients, the set of all amino acids in all TMDs of EMC clients was collected, and the fraction of such amino acids that were aliphatic, aromatic, charged, hydrophobic, or polar was calculated. The same calculations were carried out for all annotated TMDs, and the ratio of fraction in TMDs in EMC clients to fraction in all TMDs was computed for each property. To assess statistical significance, random subsets of the same number of TMDs as in the total EMC client set were drawn from the set of all TMDs and the same ratio of fractions was computed. This process was repeated for 10,000 random subsets and the fifth and 95th percentile of the 10,000 ratios produced were recorded.

## Targeted downregulation of gene expression by CRISPRi

CRISPRi HeLa cell lines were generated by transducing with pHR-SFFV-dCas9-BFP-KRAB (Addgene ID: 46911) and sorting for BFP positive cells (*Gilbert et al., 2013*). EMC2 (GAGTACGCG TCCGGGCCAA), EMC4 (GTCATTTCCGCCCTGGAAAT) and negative control Gal4-4 (GAACGAC TAGTTAGGCGTGTA) protospacers were cloned into a lentiviral expression plasmid expressing guides from a mouse-derived U6 promoter, BFP and puromycin (pU6-sgRNA EF1-Alpha-puro-T2A-BFP; Addgene ID: 60955). HeLa CRISPRi cells were transduced with guide RNA expression plasmids, selected in puromycin for 72 hr and either directly used for experiments or expanded for SILAC labeling. HeLa cell lines were confirmed by STR analysis and tested as free from mycoplasma contamination.

## Mammalian ribosome profiling

Ribosome profiling was performed for HeLa-dCas9-KRAB cells, and HeLa-dCas9-KRAB cells expressing EMC2 or Gal4-4 control guide RNAs. Cells were cultured in 15 cm plates with Dulbecco modified eagle medium (DMEM) with 10% fetal bovine serum (Gibco) until ~80% confluency. Cells were treated with 100 ug/ml CHX for 2 min and then lysed using 500 µl per polysome lysis buffer (20 mM Tris pH 7.5, 150 mM NaCl, 5 mM MgCl$_2$, 1% Triton x-100, 1 mM DTT, 8% glycerol, 100 µg/ml CHX, 24 U/ml Turbo DNase) per plate using a rubber cell scraper to facilitate lysis. Lysate was centrifuged for at 20,000 x g for 2 min at 4°C and the remaining cleared polysome-containing lysate was flash frozen by immersion in liquid nitrogen and stored at -80°C until digestion. CaCl$_2$ was added to

polysome-containing lysate to a final concentration of 5 mM and 30 µg was digested to monosomes using micrococcal nuclease (8 U/µg) for 1 hr at room temperature and the reaction was terminated by the addition of EGTA (6.25 mM). Digested lysates were equilibrated to 500 µl with polysome gradient buffer (20 mM Tris pH 7.5, 150 mM NaCl, 5 mM MgCl$_2$, 1% Triton x-100, 1 mM DTT, 100 µg/ml CHX) and loaded on top of a sucrose cushion (polysome buffer containing 1.65 M sucrose) and ultracentrifuged in a TLA-110 rotor (Beckman Coulter) for 4 hours at 4°C. The monosome-containing pellet was resuspended in 700 µl Trizol (Life Technologies). Total RNA was extracted and libraries were prepared as described for proximity-based ribosome profiling.

## Quantitative SILAC mass spectrometry

### Cell culture and harvesting

All cell culture reagents were obtained from Gibco unless otherwise stated. Cells were cultured for at least seven doublings in SILAC DMEM supplemented with 10% Dialyzed FBS, 20 Units/mL Penicilin, 20 µg/ml Streptomycin, 1 mM Sodium Pyruvate, 10%, and 2 mM L-alanyl-L-glutamine dipeptide and either; 42 mg/L $^{13}$C$_6$,$^{15}$N$_4$-L-Arginine HCl (Silantes) together with 73 mg/L $^{13}$C$_6$,$^{15}$N$_2$-L-Lysine HCl (Silantes), or 42 mg/L Arginine HCl and 73 mg/L Lysine HCl with standard isotopic constituents (Sigma). Cells were harvested by rinsing twice in ice-cold PBS excluding Calcium Chloride and Magnesium Chloride. Cells from 1 × 10 cm dish were scraped in 1 ml of ice-cold PBS and transferred to a 1.5 mL Eppendorf tube, for centrifugation at 300 x *g* at 4°C. The PBS was aspirated and cells were resuspended in lysis buffer (4% SDS, 100 mM DTT in 100 mM Tris-HCl, pH 7.6 at room temperature) and heated to 95°C for 5 min. Samples were sonicated for 15 × 30 s cycles using a Bioruptor to reduce viscosity of the lysate. Protein concentrations were determined via tryptophan assay.

### Filter-aided sample preparation (FASP)

Peptides were generated essentially as described (*Zielinska et al., 2010*). Protein lysate from Gal4 SILAC heavy-labelled cells and EMC2 or EMC4 SILAC light-labelled cells were mixed 1:1, and 100 µg of sample plus 250 µL Urea Buffer (8M urea, 100 mM Tris pH 8.5) was loaded onto Microcon 30 kDa MWCO centrifugal filters. Loaded filters were centrifuged at 10,000 x *g* at 18°C for 20 min. Filters were centrifuged a further two times with 250 µL Urea Buffer. Samples were alkylated at room temperature for 15 min by incubation with 50 mM Iodoacetamide in Urea Buffer. Samples were centrifuged a further three times for 15 min each with 150 µL Urea Buffer, before two times centrifugation with 150 µL digestion buffer (40 mM NH$_4$HCO$_3$). Finally 2 µg trypsin (Sigma) or 6.25 µg GluC (NEB) in 40 µL digestion buffer was added to the filters and incubated overnight at 37°C. Peptides were collected by centrifugation followed by a further two washes with elution buffer (1 mM CaCl2, 1 mM MnCl2, 500 mM NaCl in 20 mM TrisHCl, pH 7.3).

### Peptide purification

Peptides were acidified with 1% (v/v) TFA, and assuming 50% recovery, 20 µg peptides were loaded directly onto SDB-RPS stage tips. Stage tips were washed twice with 0.1% (v/v) TFA, and sequentially eluted with 20 µL SDB-RPS1 (100 mM Ammonium formate, 40% (v/v) Acetonitrile, 0.5% (v/v) Formic acid), followed by 20 µL SDB-RPS2 (150 mM Ammonium formate, 60% (v/v) Acetonitrile 0.5% (v/v) Formic acid), then 30 µL SDB-RPS3 (1% (v/v) TFA, 80% Acetonitrile). Tryptic peptides were dried to completion in a centrifugal vacuum concentrator (Concentrator 5301, Eppendorf), and volumes were restored to 10 µL with buffer A* (0.1% (v/v) TFA, 2% (v/v) Acetonitrile).

### Liquid chromatography coupled to tandem mass spectrometry

LC-MS/MS was performed exactly as described previously (*Itzhak et al., 2016*), with the exception that the LC was coupled to a Q Exactive HF-X Hybrid Quadropole-Orbitrap mass spectrometer, which boasts improved ion transfer (*Kelstrup et al., 2018*).

### Processing of mass spectrometry .RAW files

Mass spectrometry .RAW files were processed in MaxQuant (*Tyanova et al., 2016*; *Cox and Mann, 2008*), version 1.5.5.2. RAW files were organized into two parameter groups to separate trypsin and GluC digested peptides. For both groups, multiplicity for was set to two, with Lys8 and Arg10 selected as heavy labels, re-quantify was turned on, with matching enabled between adjacent

peptide fractions with the same enzyme. The Fasta file Homo_sapiens.GRCh38.pep.all.fa was downloaded from Ensembl.

## Acknowledgements

We gratefully acknowledge funding and support from the following institutions and foundations: MJS is a Howard Hughes Medical Institute Fellow of the Helen Hay Whitney Foundation, JSW is an Investigator of the Howard Hughes Medical Institute and, AF is a Searle Scholar and Chan-Zuckerberg Biohub investigator. This work was further supported by the Jane Coffin Childs Memorial Foundation (NTSO), the Howard Hughes Medical Institute (ALB), the Dr. Miriam and Sheldon G. Adelson Medical Research Foundation (ALB,) a Faculty Scholar grant from the Howard Hughes Medical Institute (AF), the Sandler Family Foundation through the UCSF Program for Breakthrough Biomedical Research (AF), the American Asthma Foundation (AF), the Max Planck Society for the Advancement of Science (GHHB, DNI), the German Research Foundation (Gottfried Wilhelm Leibniz Prize MA 1764/2-1 (GHHB), the louis-Jeantet Foundation (DNI), the European Research Council Synergy Grant 'ToPAG – Toxic Protein Aggregation in neurodegeneration' (ERC2012-SyG_318987-ToPAG (DNI), and the National Institutes of Health (NIH) (GM075061 - JLB, AG041826 - JSW, 1DP2GM110772-01 - AF, 8P41GM103481 and 1S10OD16229 –ALB). We thank Kendra Swain, Christopher Williams and Maya Schuldiner for their experimental and intellectual contributions. We also thank Lakshmi Miller-Vedam, Marco Jost and Marco Hein for critical reading of the manuscript, Jeffrey Quinn for producing model figures, and the rest of the Weissman lab for helpful discussions and various contributions.

## Additional information

### Funding

| Funder | Grant reference number | Author |
|---|---|---|
| Howard Hughes Medical Institute | Investigator Program | Jonathan S Weissman |
| National Institutes of Health | GM075061 | Jeffrey Brodsky |
| Helen Hay Whitney Foundation | Postdoctoral Fellowship | Matthew J Shurtleff |
| Jane Coffin Childs Memorial Fund for Medical Research | Postdoctoral Fellowship | Nicole T Schirle Oakdale |
| Sandler Foundation | Program for Breakthrough Biomedical Research | Adam Frost |
| American Asthma Foundation | | Adam Frost |
| Louis-Jeantet Foundation | | Daniel N Itzhak |
| Dr. Miriam and Sheldon G. Adelson Medical Research Foundation | | Alma L Burlingame |
| Max-Planck-Gesellschaft | | Daniel N Itzhak |
| Deutsche Forschungsgemeinschaft | Gottfried Wilhelm Leibniz Prize MA 1764/2-1 | Georg HH Borner |
| European Research Council | ERC2012-SyG_318987-ToPAG | Daniel N Itzhak |
| Howard Hughes Medical Institute | Faculty Scholar Grant | Adam Frost |
| National Institutes of Health | AG041826 | Jonathan S Weissman |
| National Institutes of Health | 1DP2GM110772-01 | Adam Frost |
| National Institutes of Health | 8P41GM103481 | Alma L Burlingame |
| National Institutes of Health | 1S10OD16229 | Alma L Burlingame |

The funders had no role in study design, data collection and interpretation, or the decision to submit the work for publication.

## Author contributions

Matthew J Shurtleff, Conceptualization, Formal analysis, Funding acquisition, Validation, Investigation, Visualization, Methodology, Writing—original draft, Writing—review and editing; Daniel N Itzhak, Conceptualization, Data curation, Formal analysis, Funding acquisition, Validation, Investigation, Visualization, Methodology, Writing—original draft, Writing—review and editing; Jeffrey A Hussmann, Conceptualization, Data curation, Software, Formal analysis, Validation, Investigation, Visualization, Methodology, Writing—original draft, Writing—review and editing; Nicole T Schirle Oakdale, Conceptualization, Funding acquisition, Validation, Investigation, Visualization, Methodology, Writing—review and editing; Elizabeth A Costa, Conceptualization, Formal analysis, Validation, Investigation, Visualization, Methodology, Writing—review and editing; Martin Jonikas, Conceptualization, Data curation, Formal analysis, Investigation, Methodology, Writing—review and editing; Jimena Weibezahn, Shruthi S Vembar, Investigation, Methodology; Katerina D Popova, Formal analysis, Validation, Investigation, Methodology, Writing—review and editing; Calvin H Jan, Data curation, Formal analysis, Investigation, Visualization, Methodology; Pavel Sinitcyn, Formal analysis, Investigation, Visualization, Methodology; Hilda Hernandez, Resources, Data curation, Formal analysis, Methodology; Jürgen Cox, Software, Supervision, Methodology; Alma L Burlingame, Resources, Funding acquisition, Methodology; Jeffrey L Brodsky, Conceptualization, Supervision, Funding acquisition, Writing—review and editing; Adam Frost, Supervision, Funding acquisition, Writing—review and editing; Georg HH Borner, Conceptualization, Supervision, Funding acquisition, Investigation, Methodology, Writing—review and editing; Jonathan S Weissman, Conceptualization, Supervision, Funding acquisition, Visualization, Writing—original draft, Writing—review and editing

## Author ORCIDs

Matthew J Shurtleff (ID) http://orcid.org/0000-0001-9846-3051
Elizabeth A Costa (ID) http://orcid.org/0000-0001-8365-0436
Pavel Sinitcyn (ID) https://orcid.org/0000-0002-2653-1702
Adam Frost (ID) https://orcid.org/0000-0003-2231-2577
Georg HH Borner (ID) https://orcid.org/0000-0002-3166-3435
Jonathan S Weissman (ID) http://orcid.org/0000-0003-2445-670X

## Decision letter and Author response

Decision letter https://doi.org/10.7554/eLife.37018.022
Author response https://doi.org/10.7554/eLife.37018.023

# Additional files

## Supplementary files

• Supplementary file 1. EMC client proteins identified by mass spectrometry. The number of transmembrane domains (TMDs), whether the protein enters the secretory pathway based on Uniprot annotation and the Log2 fold changes in protein abundance in EMC2 or EMC4 depleted cells compared to control cells expressing a non-targeting sgRNA (GAL4).
DOI: https://doi.org/10.7554/eLife.37018.012

• Supplementary file 2. Transmembrane domain annotations in yeast. A list of all transmembrane domains annotated in the yeast proteome for analyses performed in *Figure 3*. See Materials and methods for details on TMD annotation.
DOI: https://doi.org/10.7554/eLife.37018.013

• Transparent reporting form
DOI: https://doi.org/10.7554/eLife.37018.014

## Data availability

Sequencing data have been deposited in GEO under accession code GSE112891.

The following dataset was generated:

| Author(s) | Year | Dataset title | Dataset URL | Database, license, and accessibility information |
|---|---|---|---|---|
| Costa EA, Popova KD, Schirle Oakdale NT, Jan CH, Weissman JS | 2018 | The ER membrane protein complex interacts cotranslationally to enable biogenesis of multipass membrane proteins | http://www.ncbi.nlm.nih.gov/geo/query/acc.cgi?acc=GSE112891 | Publicly available at the NCBI Gene Expression Omnibus (accession no: GSE112891) |

The following previously published datasets were used:

| Author(s) | Year | Dataset title | Dataset URL | Database, license, and accessibility information |
|---|---|---|---|---|
| Aviram N, Ast T, Costa EA, Arakel EC, Chuartzman SG, Jan CH, Haßdenteufel S, Dudek J, Jung M, Schorr S, Zimmermann R, Schwappach B, Weissman JS, Schuldiner M | 2016 | The SND proteins constitute an alternative targeting route to the endoplasmic reticulum | http://www.ncbi.nlm.nih.gov/geo/query/acc.cgi?acc=GSE85686 | Publicly available at the NCBI Gene Expression Omnibus (accession no: GSE85686) |
| Jan CH, Williams CC, Weissman JS | 2014 | Principles of ER Co-Translational Translocation Revealed by Proximity-Specific Ribosome Profiling | http://www.ncbi.nlm.nih.gov/geo/query/acc.cgi?acc=GSE61012 | Publicly available at the NCBI Gene Expression Omnibus (accession no: GSE61012) |

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
