## [Decision Letter]

Thank you for submitting your article "The ER membrane protein complex interacts cotranslationally to enable biogenesis of multipass membrane proteins" for consideration by *eLife*. Your article has been favorably evaluated by a Vivek Malhotra (Senior Editor) and three reviewers, one of whom is a member of our Board of Reviewing Editors. The following individuals involved in review of your submission have agreed to reveal their identity: Gunnar von Heijne (Reviewer #2); Ari Helenius (Reviewer #3). The complete, un-redacted comments of all three are posted below.

We have discussed the reviews with one another and have drafted this decision to help you prepare a revised submission.

Summary:

This study addresses an important question related to multipass membrane protein maturation and quality control in the ER. Although it has been previously shown that the EMC complexes interact with such proteins, the mechanisms and detailed functions have not been analyzed. Here, elegant and extensive studies reveal not only a spectrum of client proteins in yeast and human cells but also describe important aspects related to molecular mechanism. The EMC complex associates with cargo molecules already co-translationally following the synthesis of clusters of transmembrane domains (TMDs) with specific properties shared by many transporters.

The findings in this paper are conceptually important and they are of wide-reaching significance. The overall mechanism with a direct interaction of nascent-growing client polypeptides with EMC modulated by client-specific co-chaperones and general folding factors is intriguing. The experiments are elegant, the documentation convincing, the text well-written, and the Discussion interesting and balanced. Therefore, we find the paper acceptable for publication, subject to some revision.

Essential revisions:

Reviewer's 1 and 2 raised concerns in regards to Figure 6, addressing the role of the EMC in biogenesis of the multipass protein BCAP31. Both the strength of the evidence that EMC deletion affects the fate of BCAP31 directly and the strength of the evidence that the rescued BCAP31 is a membrane associated species are questioned. Therefore we feel that these limitations need to be addressed in the revised manuscript. As options, we suggest eliminating the figure altogether, or if you have some other more creative approach to the problem, please feel free to incorporate it into the revised version.

Reviewer #1:

This paper significantly advances the notion that the EMC is implicated in the biogenesis of multipass transmembrane proteins whose mature folded structure places heavy constrains on their stability during biogenesis. While this idea has been in circulation for some time; essentially since the discovery of the EMC, the experimental support for this notion has to date not been strong.

Proximity-based profiling of the mRNA associated with ribosomes found in proximity to the EMC reveals these to be enriched for sequences encoding the aforementioned class of special multi-pass TM proteins. The resolution of the proximity-based profiling is such that one may conclude that the ribosome-nascent chain complex encoding multipass transmembrane proteins whose mature folded structure places heavy constrains on their stability during biogenesis associate with the EMC around the time the challenging segment is synthesized and remains associated with the EMC until completion of synthesis. These findings, reported on in Figures 3 and 4 are the hard core of the paper, but they are nicely buttressed by correlative findings concerning the phenotype of EMC loss in yeast and mammalian cells as well as information on the protein contingents of the EMC.

This reviewer, who is not an expert in the area, found this paper easy to read and was persuaded by the novelty and importance of the findings in the paper as is.

Reviewer #2:

The EMC has recently emerged as an important ER complex, functioning in the biogenesis of membrane proteins, whose functions and mode of operation are largely uncharacterized. In this manuscript, Shurtleff et al. use several high-throughput techniques to better define the complex components, its clientele and time of engagement by the EMC. The use of MS-based approaches helps revealing which proteins interact with the EMC in yeast and which putative client proteins get destabilized by its depletion in human cells. Most convincingly, proximity ribosome profiling in yeast provides clear evidence for cotranslational interaction of the EMC with its clients and helps in directly identifying them. The work reveals a major role in cotranslational chaperoning of multi-spanning membrane proteins that likely continues also post-translationally. Overall this well written manuscript describes work that represents an important step in our understanding of this important complex. As such, I recommend its publication without delay.

1) In the last section of the paper (Figure 6), the authors try to better characterize the mechanism of the EMC by zeroing in on BCAP31 as a 'model' EMC substrate. This part feels premature, primarily because it is not completely clear if BCAP31 represents a true substrate. Its degradation upon EMC depletion might suggest it to be a substrate, but this can also be a secondary effect. The authors themselves see some probable secondarily-degraded proteins, containing no TM, in their analysis of the human proteome. While the authors rule out a secondary effect of FDFT1 in BCAP31 degradation, it could still be a secondary effect that depends on another protein. In any case this part does not provide much mechanistic insight. Since most EMC substrates have multiple TMs and are targeted by the SRP – it is pretty clear that even in the absence of the EMC they will be inserted (at least some of their TMs) and not completely cytosolic. I therefore believe that this part can be omitted from the paper without compromising the rest of the work reported in the other sections.

2) The parts that discuss the post-translational interaction of the EMC with its clients need to be written with more caution since there is insufficient information about this in the work reported, except for a few full-length protein clients identified in Figure 1A.

Reviewer #3:

This study addresses an important question related to multipass membrane protein maturation and quality control in the ER. Although it has been previously shown that the EMC complexes interact with such proteins, the mechanisms and detailed functions have not been analyzed. Here, elegant and extensive studies reveal not only a spectrum of client proteins in yeast and human cells but also describe important aspects related to molecular mechanism. The EMC complex associates with cargo molecules already co-translationally following the synthesis of clusters of transmembrane domains (TMDs) with specific properties shared by many transporters.

The findings in this paper are conceptually important and they are of wide-reaching significance. The overall mechanism with a direct interaction of nascent-growing client polypeptides with EMC modulated by client-specific co-chaperones and general folding factors is intriguing. The experiments are elegant, the documentation convincing, the text well-written, and the Discussion interesting and balanced. Therefore, I find the paper fully acceptable for publication.

There is one interesting aspect that could be discussed more deeply. It has to do with client specificity. A large family of multipass-proteins, the channel proteins, do not seem to be represented. Why is that? Channel proteins also have TMDs that contain charged residues but they are oligomers. Could the EMC association correlate with the oligomeric state of the mature clients?

---

## [Author Response]

Reviewer #2:[…] 1) In the last section of the paper (Figure 6), the authors try to better characterize the mechanism of the EMC by zeroing in on BCAP31 as a 'model' EMC substrate. This part feels premature, primarily because it is not completely clear if BCAP31 represents a true substrate. Its degradation upon EMC depletion might suggest it to be a substrate, but this can also be a secondary effect. The authors themselves see some probable secondarily-degraded proteins, containing no TM, in their analysis of the human proteome. While the authors rule out a secondary effect of FDFT1 in BCAP31 degradation, it could still be a secondary effect that depends on another protein. In any case this part does not provide much mechanistic insight. Since most EMC substrates have multiple TMs and are targeted by the SRP – it is pretty clear that even in the absence of the EMC they will be inserted (at least some of their TMs) and not completely cytosolic. I therefore believe that this part can be omitted from the paper without compromising the rest of the work reported in the other sections.

We agree with the reviewers that the strength of the paper lies in the global analyses and that we cannot, at this stage, exclude the formal possibility that decreased BCAP31 abundance could be secondary to another depleted client (we did of course explicitly exclude this possibility for what we thought was the most immediate concern in this regard i.e. that loss of BCAP31 is a secondary consequence of FDFT1. Per the reviewers’ suggestions we have removed the BCAP31 data from this figure and hope to present this in the future as part of an expanded mechanistic study.

2) The parts that discuss the post-translational interaction of the EMC with its clients need to be written with more caution since there is insufficient information about this in the work reported, except for a few full-length protein clients identified in Figure 1A.

In addition to the full-length clients shown in Figure 1A, we also observed peptides spanning the full-length of client proteins in pulldown mass spec analyses in Figure 1B and Figure 2B. However, we agree that, given the limited number of clients detected using these approaches, that post-translational interaction between the EMC and clients as a general rule remains to be more fully explored. In the revised version we have toned down this claim throughout.

For example: “The presence of full-length multipass membrane proteins in our pulldown analyses in yeast (Figures 1 and 2) indicate that the EMC can engage clients cotranslationally and remain bound after completion of protein synthesis for at least a subset of client proteins.”

Reviewer #3:[…] There is one interesting aspect that could be discussed more deeply. It has to do with client specificity. A large family of multipass-proteins, the channel proteins, do not seem to be represented. Why is that? Channel proteins also have TMDs that contain charged residues but they are oligomers. Could the EMC association correlate with the oligomeric state of the mature clients?

The reviewer raises as interesting point. Indeed, we do not observe oligomeric channel proteins as being among the list of proteins of decreased abundance as a upon EMC depletion in HeLa cells. However, the EMC has previously been implicated in the biosynthesis of such channel proteins in *C. elegans* (acetylcholine receptors and GABA receptor) (Richard, 2013). These proteins however are not expressed in HeLa cells. Based on this observation, we suspect that some, but not all, oligomeric channel proteins are EMC clients in specific cell types, however this will require further examination in future studies.

To clarify this point for the reader, we have added the following sentence to the Discussion:

“While our studies in HeLa cells primarily identified solute carriers and transmembrane ATPases as clients, the EMC was previously implicated in the biogenesis of cell-type specific multimeric channel proteins that also contain TMDs with charged residues (Richard et al., 2013).”